# Defining a Characteristic Gene Expression Set Responsible for Cancer Stem Cell-Like Features in a Sub-Population of Ewing Sarcoma Cells CADO-ES1

**DOI:** 10.3390/ijms19123908

**Published:** 2018-12-06

**Authors:** Marc Hotfilder, Nikhil Mallela, Jochen Seggewiß, Uta Dirksen, Eberhard Korsching

**Affiliations:** 1Department of Pediatric Hematology and Oncology, University Hospital Münster, 48149 Münster, Germany; marc.hotfilder@ukmuenster.de; 2Institute of Bioinformatics, Faculty of Medicine, University of Münster, 48149 Münster, Germany; nikhil.vinod2006@gmail.com; 3Institute of Human Genetics, Faculty of Medicine, University of Münster, 48149 Münster, Germany; jochen.seggewiss@ukmuenster.de; 4University Hospital Essen, Pediatrics III, Hematology and Oncology, West German Cancer Centre, 45147 Essen, Germany; uta.dirksen@uk-essen.de

**Keywords:** Ewing sarcoma, side population, tumor driver cells, cancer stem cells, mesenchymal stem cells, gene expression, AP-1, APC/c

## Abstract

One of the still open questions in Ewing sarcoma, a rare bone tumor with weak therapeutic options, is to identify the tumor-driving cell (sub) population and to understand the specifics in the biological network of these cells. This basic scientific insight might foster the development of more specific therapeutic target patterns. The experimental approach is based on a side population (SP) of Ewing cells, based on the model cell line CADO-ES1. The SP is established by flow cytometry and defined by the idea that tumor stem-like cells can be identified by the time-course in clearing a given artificial dye. The SP was characterized by a higher colony forming activity, by a higher differentiation potential, by higher resistance to cytotoxic drugs, and by morphology. Several SP and non-SP cell fractions and bone marrow-derived mesenchymal stem cell reference were analyzed by short read sequencing of the full transcriptome. The double-differential analysis leads to an altered expression structure of SP cells centered around the AP-1 and APC/c complex. The SP cells share only a limited proportion of the full mesenchymal stem cell stemness set of genes. This is in line with the expectation that tumor stem-like cells share only a limited subset of stemness features which are relevant for tumor survival.

## 1. Introduction

Ewing sarcoma is the second most frequent malignant bone cancer after osteosarcoma and was first described by James Ewing in 1921 [1]. Ewing sarcoma is rare, highly aggressive and most frequently affects children and young adults with an average incidence of about one to three cases per million people per year [2]. Ewing sarcoma shows a slight predominance in males in comparison to females (55/45 ratio) and its incidence is nine times greater in Caucasians than in African-Americans [3]. Histochemically, Ewing sarcoma is characterized as a small round blue-cell undifferentiated aggressive tumor of the bone and occasionally soft tissues with a marked propensity for dissemination [4,5]. Despite significant progress in treating Ewing sarcoma over the last years, the prognosis still remains poor with an event-free survival of less than 20% [6].

On the molecular level, Ewing sarcoma is characterized by aberrant chromosomal translocations usually involving EWSR1 (22q12), a gene belonging to the TET protein family, and one of the genes from ETS family of transcription factors. The FLI1 (11q24) [7] and ERG (21q22) [8] genes from the ETS family are the most common translocation partners. EWSR1-FLI1 t(11;22)(q24;q12) and EWSR1-ERG t(21;22)(q22;q12) fusions accounts for 85% and 5–10% of the cases respectively [9]. The chimeric fusion proteins are produced when the N-terminal transactivation domain of EWSR1 combines with the C-terminal DNA binding domain of FLI-1 or ERG [7]. Both FLI1 and ERG share 68% amino acid identity over their entire peptide sequence and 98% (83/85) amino acid identity in their ETS-binding domains [10]. The other less frequent ETS-family translocation partners include ETV1 (7p22), ETV4/E1AF (17q12) and FEV (2q33) genes [11]. These mentioned main features of the Ewing sarcoma form the mature appearance of the tumor when diagnosed in the clinic.

However, there is more complexity in a mature tumor. It is now common sense that the tumor is also composed on a multitude of cell sub-populations creating the diversity and stability of the progressing tumor [12]. The majority of the tumor cells will not have the potential to progress or maintain the general viability of the tumor. This small cell fraction is generally termed cancer stem cell (CSC). The CSCs possess characteristics associated with normal stem cells such as their ability to give rise to several cell types on a wide spectrum of differentiation stages and various proliferative capacities. Those with the ability for self-renewal become CSCs again, which were described among others by John Dick in acute myeloid leukemia [13].

In Ewing sarcoma, the multipotent mesenchymal stem cells (MSCs) can be seen as progenitor cells of the putative Ewing sarcoma CSCs [14,15]. MSCs are the nonhematopoietic multipotent stem cells of the bone marrow [16]. MSCs give rise to several differentiated cells including osteoblasts, osteocytes, adipocytes, chondrocytes, stromal cells, and myogenic precursors such as cardiac muscles, skeletal muscles and smooth muscles. Experiments involving the knockdown of Ewing sarcoma fusion genes in Ewing sarcoma cell lines have resulted in MSC-like gene expression and phenotype [14,17,18]. Other experiments involving the expression of Ewing sarcoma fusion genes in MSCs have induced Ewing sarcoma-like malignant transformation [18,19]. So characterizing the molecular gene expression networks of MSCs and Ewing CSCs in more detail would give more insight into the dependency nature of master gene regulators.

The presented study is comparing the gene expression pattern of bone marrow MSCs with an Ewing sarcoma ‘side population’ (SP) based on the CADO-ES1 cell line owning CSC properties, to identify common regulatory mechanisms contributing to the CSC characteristics of Ewing sarcoma.

The SP termed cell fraction is defined by flowcytometry. These cells have a higher drug efflux capability for anti-mitotic drugs or artificial dyes, as well as a higher ability to transdifferentiate. Analogous to the normal stem cells, the cancer stems cells have also been identified to harbor tumorigenic SP population [20]. The SPs are capable of sustained expansion, generating both SP and non-SP progeny clearly pointing toward the hierarchical model of cancer stem cell theory [21], a model that suggests that only a sub-population of the cancer stem cells have the ability to drive the progression of cancer. These cells express transporter genes belonging to the ABC transporter family, thereby enhancing their capacity to expel foreign substances, resulting in better survival [20].

The objectives of the study are (I) to isolate SP and non-SP cell types from the Ewing model cell line CADO-ES1, to characterize some typical stemness-like features and (II) to identify active regulatory mechanisms that are specifically responsible for the CSC properties of the SP cells in Ewing sarcoma.

## 2. Results

### 2.1. Defining a CADO-ES1 Side Population

Many articles have now been published which support the clonal nature of human cancers (e.g., [22,23]) and the putative roles of those cells having different capabilities, but acting in a systemic manner. The isolation of the CSC-like cells were done according to the established procedures of Goodell [24]. The established SP cells, a fraction of 1 to 3% of the whole cell line population, exhibit features like: The ability to form more colonies in an anchorage-independent growth assay compared to the non-SP cell population (Figure 1A, 1D-left one), to differentiate to a higher extent into adipogenic cells compared to non-SP cells (Figure 1B: SP left, non-SP right) and equally to MSC (not shown), are slightly smaller (Figure 1C), and are more resistant to treatment with certain cytotoxic drugs (Figure 1D middle, right).

All the analyzed features show differences between the SP and non-SP population and indicate some stemness characteristics of the SP population.

### 2.2. Double Differential Experimental Design and Short Read Sequencing

The experimental design is composed of CADO-ES1 Ewing tumor cells and bone marrow-derived MSCs. The CADO-ES1 cell line is subdivided as described before into SP, non-SP, and two further technical controls: Not sorted but stained and sorted and not stained. The later controls were designed for intercepting those differential genes which are due to staining or sorting stress. However, it turned out, that on the level of three biological replicates for each condition, the variability will dilute the effects of the main case control design. Therefore, these controls are not considered but are part of the published, more comprehensive, main data set. The overall analysis workflow is illustrated in Figure 2.

From all samples, total RNA was isolated and the ribosomal RNA proportion was removed. About 200 million short reads were generated per sample with a standard Life Technology workflow on Solid4/5500XL sequencers. The reads were checked for quality with the FastQC program. The adapters, if there, were removed by Cutadapt and alignment performed by Tophat. The on average 30 million aligned high quality reads per sample were quantified on a per gene basis by HTSeq.

### 2.3. The Identification of Tumor Driver Gene-Sets

The identification of differentially expressed genes (DEG) is based on two linked differential analyses and two intersections. The used subsets are SET-1: Difference of the side population cells to the non-SP population, SET-2: Intersection of up-regulated SET-1 genes with up-regulated MSC cells and SET-3: The intersection of down-regulated SET-1 genes with down-regulated MSC cells.

The idea behind SET-1 is that these differential genes might help us to understand the distinguishing features of SP cells which are hypothesized to be tumor-driver cells owning stem cell-like properties. The hypothesis in deriving SET-2 and SET-3 is that these genes might help to understand the common regulatory mechanisms between MSCs and SPs, where MSC might be the proposed cell of origin for Ewing sarcoma.

The transcriptomic sequencing data sets of SP, non-SP and MSC after the alignment to the reference genome HG38 patch 10 have an average of 26x coverage. The differential analysis of SP versus non-SP cells, according to the DESeq2 standard protocol, resulted in a total of 312 transcripts (SET-1) where 215 genes are up-regulated and 97 genes are down-regulated (File_S1_312_SET_1). The second differential analysis between SP and MSC cells resulted in a differential set of around 10,000 genes owning 5% more up-regulated genes than down-regulated genes. This excessive amount of differential genes is a common phenomenon comparing stem cells and differentiated cells and does not hamper the double differential approach. The alpha threshold was chosen at a more permissive value of 0.1 because the main application objective is to use gene set enrichment procedures and set size is critical in this respect.

The intersection of the up-regulated genes of SET-1 and the up-regulated genes of SP versus MSC resulted in 42 transcripts (SET-2) that are commonly up-regulated in SP and MSC but are down-regulated in the non-SP. The analogous intersection, except for the down-regulated genes, resulted in 46 transcripts (SET-3).

The expression data of the up-regulated 42 genes (SET-2) across all replicates and all conditions can be seen in Figure 3. Three coherent expression groups can be identified, illustrated by the cluster tree on the left. In two cluster groups, the average expression activity is increasing from the non-SP over SP to the MSC samples, while in the remaining group the SP samples have a stronger activity window. The fold change of the regulation is not balanced.

The down-regulated genes (SET-3) do not really show a well-defined cluster (Appendix A). The difference between the MSCs and the group of SPs/non-SPs is remarkably clear. Only sparse differences exist between SP and non-SP. So the SET-3 gene expression characteristic is in a certain way unrelated to SET-2.

An overview of all the result sets by their gene symbols is given in ‘File_S2_overview_sets’.

### 2.4. Functional and Pathway Enrichment Analysis

The 312 DEGs in the SET-1 are first classified according to gene function using the Gene Functional Classification tool of DAVID knowledge-base. According to this analysis, the genes are observed to have clustered into six prominent clusters which can be summarized as phosphoprotein cluster, histone cluster, cell cycle genes cluster, protein kinase cluster, kinesin family cluster, and zinc finger protein cluster (File_S3_DAVID_312_SET_1). Similar analysis of the SET-2 revealed one prominent cluster, the histone cluster (File_S4_DAVID_42up_SET_2).

GO functional analysis has been performed using the BiNGO plugin of Cytoscape with the up-regulated genes of the SET-1. The biological processes terms such as cell cycle phase, M phase, and organelle organization were found to be significantly enriched (Table 1 and Appendix A). Pathway analysis using CPDB with the up-regulated genes of SET-1 revealed cell cycle, cell division, mitotic prophase, and cellular senescence as some of the significant pathways (Table 2). The pathway analysis with SET-2 genes revealed cellular senescence, oxidative stress-induced senescence, cellular responses to stress, and AP-1 transcription factor network pathways as significant (Table 2).

The number of APC/c complex-associated factors in SET-1 seems to be small (3%) but they represent 39% of the known APC/c members (Figure 4). This is an important change.

For the down-regulated fraction SET-3, the GO functional classification and the pathway analysis revealed, on the used test constraints and FDR corrected *p* values, no significant pathways and GO processes (File_S5_DAVID_46down_SET_3). Because these genes are less informative concerning enrichment procedures, the further enrichment analyses were performed and reported only for the up-regulated genes.

### 2.5. Identification of Oncogenes and Tumor Suppressor Genes

According to the annotation, 43 genes of 312 DEGs (SET-1) were identified as tumor-associated genes (File_S2_overview_sets). These known oncogenes are not forming any cluster in the Gene Functional Classification tool of DAVID (File_S6_DAVID_43_oncogenes). Among the 35 up-regulated and annotated genes, 21 are oncogenes (KIF14, ID2, COPS3, UBE2C, SGK1, E2F5, ATF1, FAM72A, PBK, FAM83D, CDC25C, CDK1, MYC, CXCL1, CCNB2, CDKN3, ID1, AURKA, CCNB1, FOS, JUN).

There are a further eight tumor-suppressor genes (DLEU2, CDKN2C, SPRY4, UBE2QL1, LIN9, TFPI2, LRIG3, DUSP1) and six genes serve as both oncogenes and tumor-suppressor genes (FOXO1, CAV1, KLF6, CDK6, PLK1, CTGF). Among the eight down-regulated genes, one is an oncogene (NEAT1), six are tumor-suppressor genes (ASS1, PTPRD, ISG15, TGFBI, SELENBP1, MEG3) and one gene serves as both an oncogene and tumor-suppressor gene (CDH17). An overview on the distribution can be found in Appendix A.

In order to observe the extent of the oncogene presence in the top enriched functional processes and pathways, the genes of the functional enrichment results have also been annotated with an ‘oncogene’ or ‘tumor-suppressor gene’ tag (Appendix A). This subset of genes again points to similar cellular processes as found during the analysis of the whole sets.

### 2.6. Identifying Epigenetic Modifier

The up-regulated SET-1 gene candidates as well as the down-regulated genes, represent a gene pool which might show an epigenetic modifier. For this purpose, the epigenetic modifiers of the curated dbEM database [25] were manually exported into a list. This list of gene symbols was imported into the R platform and intersected with the gene symbol identifier of SET-1 and also SET-2. Only in SET-1 an overlap to dbEM candidates was found: HDAC9, a histone deacetylase.

### 2.7. The Protein-Protein Interaction (PPI) Network Analysis Is Supporting the Annotation Derived Information

To exploit the existing knowledge on protein interactions and to get insight into putative interaction networks, the 312 SET-1 DEGs were supplied as an input to the STRING database. A PPI network of 182 gene products (157 up-regulated, 25 down-regulated) with 2056 interactions was retrieved. The network was then imported into Cytoscape and the network statistics were calculated to identify highly connected nodes (so called ‘hubs’) characterizing the network topology which implicitly is pointing to the biological function. TOP2A (degree = 87), CDK1 (degree = 82), CCNB1 (degree = 80), CENPA (degree = 74), and CCNA2 (degree = 68) are the top five genes with the highest degree of connectivity in the complete network (Appendix A). CDK1 and CCNB1 are also part of the oncogene group. The network can be inspected online [26] or offline (File_S7_network).

Taking the SET-2 genes alone for constructing the PPI network reveals again the scenario around AP-1 and the histone cluster (Figure 5).

jActiveModules [27] was employed to analyze the substructure of the whole PPI network. The objective of this analysis is to unfold complex networks into subunits, which again own some functional topic. This is realized by finding synergistic changes in expression according to selected conditions. The utilized *p* values for this analysis were chosen from the FDR adjusted *p* values of the DEGs of the SP and non-SP comparison (SET-1). The search space was limited to display five significant subnetworks (Appendix A). The two highest scoring subnetworks are seen in Figure 6A,B. The first highest scoring subnetwork has 6 nodes and 11 interactions, while the second one has 36 nodes and 216 interactions. The first network is pointing to the activity of the AP-1 complex and the second is situated again close to cell cycle progression and cellular transformation.

### 2.8. Condensing Information on Networks and Pathways by Considering on Protein Domains and Protein Complexes

The generated differential gene sets were analyzed up to now concerning their enrichment in categories and functional roles. Additionally, now PPI information from the STRING database was used to model the network structure in between the molecular factors. The resulting network and sub-networks support the functional analysis.

These two approaches can be further complemented by additionally finding an enrichment on the level of functional protein domains as well as on known protein complexes.

The domain view is subdividing the view on genes/proteins in a functional way. In Table 3, the results are shown. Again, many functions are pointing to transcriptional control (bZIP family, kinases, histones) as well as to general control of regulatory procedures (Insulin domain—growth factor, cyclin—cell cycle, von Willebrand factor).

In SET-2, additional protein stability/growth factor activity shows up (Cystein-knot). Up to 10 percent of the study factors per domain and overall approximately 23 percent of the study factors are represented. These results are again coherent with the GO-based results from the beginning.

The protein complex view (Table 4) has a slightly different logic. Here, it is a fixed set size rather a variable one. However, we see that nearly all selected candidates go into the presented complex motives. Again, the same molecular evidences show up in a focused manner for SET-1 and SET-2: AP-1, CDC2 centered activity.

## 3. Discussion

Whether the Ewing sarcoma fusion gene marks the origin of the Ewing sarcoma or is acquired during tumor progression is still not known. Nevertheless, the tumor phenotype is definitely formed by this fusion protein type and the physiological cellular processes are altered remarkably [28]. However, the complexity is still higher. It is now established knowledge that tumors are not only based on one tumor clone but instead are composed in a multi-clonal way [29,30]. Only a sub-proportion with stem cell-like features is well suited to fulfill the maintainer and pathfinder role for the whole tumor [31]. To know more on this sub-population will benefit anti-tumor strategies as well as therapeutic methods.

This study established a SP of Ewing sarcoma cells, characterized some features of the biology and analyzed the biological networks of this sub-population in comparison to the whole population of tumor cells and a MSC reference population.

The SP are able to show in a colony-forming assay that they are able to create spheres to an extent which is characteristic for CSC-like undifferentiated cell populations [32], despite it having some limitations [33]. The differentiation potential is higher than the main tumor population, and their morphology is less differentiated and more round-shaped in contrast to the main population of cells which are more flat shaped (adhering). The SP cells tolerate higher doses of cytotoxic drugs which might be due to a faster clearance of these contaminants by typical members of the ABC transporter family [34]. All together, these still limited features are good indicators that the SP population might be, or at least be part of, a CSC population in Ewing sarcoma.

Beyond this phenotypic characterization, the SP shows also an altered pathway situation. The linked double differential case control design of SP versus non-SP and MSC versus non-SP (Figure 2) reveals some of the CSC-like properties. Looking at the details around SET-1, it can be observed that the number of up-regulated genes is twofold above those which are down-regulated. This indicates a higher overall activity of tumor-driving cells in contrast to more differentiated cancer cells; a phenomenon which is already observed by many researchers comparing cancer and normal cells e.g., [35].

The first functional classification in DAVID resulted in six functional clusters in the case of SET-1. So we see a very distinct and focused functional environment centered on cell cycle activity and regulatory mechanisms (kinases, histones) as the GO terms indicate. The functional classification of the subset (SET-2) narrowed down to the histone cluster and revealed a lot of key player like JUN, FOS which link to the AP-1 complex and transcriptional control [36,37,38].

The results of both functional and pathway analysis with the SET-1 genes, point additionally towards the cell cycle regulators as being the governing factors in steering the activity of the SP cells in contrast to the non-SP cells. The enrichment results indicate the progression of cancer mediated through the cell cycle regulators [39]. In contrast, none of the enrichment results obtained with the SET-2 have revealed enriched cell cycle regulating pathways but instead pathways related more to the stress response.

Although the GO term cell cycle gives us the information about the active set of genes driving the SP cells, it does not provide us with the required resolution with regard to the specific candidates responsible for the SP phenotype. In order to have a deeper understanding, enrichment analyses were conducted in terms of protein complexes to specifically identify those cell cycle regulators which are playing a key role in progression and stemness properties of the SP cells.

Agreeing with the gene functional classification, the protein domain enrichment analysis identified bZIP transcription factors as one of the significantly enriched domains in our DEG list (SET-1 and SET-2). bZIP transcription factors (basic region leucine zipper) play a central role in the regulation of gene expression by extracellular signals. The bZIP transcription factors have a dimerization domain and a DNA binding domain. Activator protein 1 (AP-1) is a transcription factor belonging to a family of bZIP proteins because they dimerize through a leucine-zipper motif and contain a basic domain for interaction with the DNA backbone. AP-1 is a dimer of two proteins from the Fos family (c-Fos, Fra-1, Fra-2, and FosB) and Jun family (c-Jun, JunB, and JunD) of transcription factors. In addition, members of the ATF/CREB family can replace one of the Fos or Jun proteins in the dimer [36,40,41].

The protein complex-based enrichment analysis (cf. Table 4) with both SET-1 and SET-2 has resulted in identifying AP-1 as the most enriched protein complex having JUN, FOS, FOSB, FOSL2 as the overlapping members from our list of DEGs, although JUN:FOS are the most prominently observed AP-1 dimers. Notably, the subnetwork analysis has also identified JUN and FOS among their most active subnetworks (cf. Figure 6). AP-1 controls a number of cellular processes including differentiation, proliferation and apoptosis [42]. However, Eferl and Wagner [36] note that AP-1 can be a double-edged sword in tumorigenesis, where the property of AP-1 proteins being oncogenic or anti-oncogenic depends on the combination of the dimers involved in forming the AP-1 complex. AP-1 transcription factor complexes formed by the members of the JUN and ATF family such as JUN:ATF2 are also seen to play a role in tumor formation [43]. Our analysis has shown a significant up-regulation of ATF1 in the SP cells which is not known to form a dimer with JUN [36]. However, ATF1 influences downstream target genes related to growth and survival. Its phosphorylation enhances its transactivation and transcriptional activities and enhances cell transformation.

Although JUN is often seen to be up-regulated in many cancers due to the activation of upstream oncogenes RAS, BRAF and EGFR, these genes were not found to be up-regulated in our data set. Our observations in regard to the AP-1 transcription factor and the expression of its associated oncogenes do not reliably support the proposed hypothesis if AP-1 is playing any role in the tumorigenic transformation of the cell. Nevertheless, the up-regulation of AP-1-mediated gene regulation might be playing a role in adding the stemness properties to the SP cells.

Yamashita and McCauley [44] show that the level of both AP-1 subunits (Jun and Fos) are expressed above the basal levels during cell division. The higher expression of AP-1 transcription factor in SPs and MSCs suggests a higher rate of cell division in comparison to the nonSPs. Furthermore, having parallels to MSCs in terms of a higher expression of AP-1 could be having a contributing role in the cancer stem cell property of the SP cells.

Anaphase Promoting Complex cyclosome (APC/c) is a multi-subunit E3 ubiquitin ligase enzyme and is one of the major driving forces playing a pivotal role in cellular process such as cell migration, proliferation, differentiation, senescence, apoptosis, cell cycle progression, and DNA damage repair [45,46]. APC/c here forms two sub-complexes, APC/c-CDC20 and APC/c-CDH1. The recent studies have shown that CDH1 is functioning as a tumor suppressor whereas CDC20 may function as an oncogene to promote the development and progression of human cancers [47]. CDC20 and CDH1 are the substrate-recruiting modules that activate the APC/c complex and drive the cell cycle progression at several stages of the cell cycle process [48].

Apart from the cell cycle control, it was also proved that CDC20 plays an important role in the development of human cancers [49]. CDC20 targets several key substrates for degradation to govern cell cycle progression [47]. Among those key substrates, CCNB1 (Cyclin B1), CCNA2 (Cyclin A) and NEK2 are seen to be up-regulated in the SP cells of our data set. Studies have shown an APC/c-independent activity of CDC20 in regulating gene transcription [50].

In this regard, the over expression of the tumor suppressor protein p53 is bound to the down-regulation of CDC20 and conversely, siRNA silencing of p53 is demonstrated to induce the expression of CDC20 [51]. Interestingly, the expression of p53 is undetected in the SP cells which probably might explain the increased expression of CDC20. Phosphorylation by CDK (cyclin-dependent kinase) and dephosphorylation by protein phosphatase 2A is critical for the activation of APC/c-CDC20 [52].

In our data set, CDKs are observed to be up-regulated in SPs. The pathway involving the APC/c activation and APC/c-CDC20-mediated degradation of mitotic proteins involves 32 gene members, of which 9 genes are seen to be upregulated in our DEG set (SET-1). The nine genes include PLK1, CDK1, MAD2L1, BUB1B, UBE2C, CCNA2, CDC20, NEK2, and CCNB1. This might indicate a relevant role of the APC/c-CDC20 sub-complex–mediated cell cycle control in SP.

CDC20 is frequently seen over-expressed in several cancers such as breast cancer [53], cervical cancer [54], glioblastomas [55], ovarian cancer [56] and others. Interestingly, Kato et al. [57] show that the higher expression of CDC20 is associated with males. This correlates to the observation that Ewing sarcoma is slightly more frequent in males as compared to females [3]. Although CDC20 is seen to be over-expressed in several cancers, the role of its over-expression has not yet been discussed in Ewing sarcoma. Whether the over-expression of CDC20 is playing a pivotal role in Ewing sarcoma cancer progression and if CDC20 is contributing to the cancer stem cell property of the Ewing sarcoma cells is an open question.

So, what is known on AP-1 and APC/c action and the stemness-like features, beyond the mechanistic details reported and discussed so far?

The link between MSCs as cells of origin and Ewing sarcoma was exemplified by a publication by Tirode et al. [14]. The recent publication of Satterfield et al. [58] is pointing towards a miR-130B interaction with the AP-1 complex forming an oncogenic feed-forward loop. An older study from Whitehurst et al. [59] discuesses an APC/c-associated EWSR1 fusion protein co-expression with RASSF1A, but the corresponding gene was not observed in our, specifically to stemness features adjusted, differential analysis.

Beyond Ewing sarcoma, the AP-1 complex is perfectly covered in the review of Alonso et al. [38], demonstrating the relevance in lymphoma. AP-1 is also showing up in Kaposi’s sarcoma-associated herpesvirus [60] and adult T-cell leukemia [61,62,63]. A nice AP-1-associated enhancer regulation motif based on FOXF1 [64] might also be a model for the family member FOXO1, which is exposed in our study and is associated with Rhabdomyosarcoma. A publication of Lopez-Bergami et al. [65] gives a broader overview on AP-1 pathways in cancer and has a nice crosslink in his Table 2 ‘ATF2 transcriptional targets’, the cell cycle-associated cyclin CCND1. The cyclins/kinases showing up in our study (CCNB1, CCNB2, CDK1, and CDK6) and the ATF1 family member might form a corresponding dependency concept with a similar characteristics. Newton et al. [66] presents insight into upstream pathways of AP-1, drafting a model for the discussed candidates.

The APC/c environment, perfectly presented by Alfieri et al. [67] and with a stronger pathway focus and cancer relevance in Zhou et al. [68], exposes, by comparing SET-1 candidates with factors mentioned in those two publications, that mainly G1/S phase alterations might be enhanced by the SP cells. It can be assumed that hubs like APC/c are accepting, presumably by different specificity, several combinations of family members, thereby dispatching multiple pathways in parallel and forming an alternate network state. Additional evidence is given by Harkness [69] who again shows a link to FOXO1 in our study, while Nicolau-Neto et al. [70] shows a link to the UBE2C group in our study (UBE2C, UBE2T, UBE2QL1) and Jia et al. [71] to the BUB family in our study (BUB, BUB1B).

Of course the gene level is not the sole level of action. Several other regulatory levels like the already mentioned microRNA miR-130B or epigenetic alterations of the transcriptional control might influence cellular expression. The dbEM database [25] is collecting 167 well described epigenetic modifiers which are known to be involved in many cellular regulation circuits. The intersection of these epigenetic modifiers with the SET-1 candidates revealed the molecular factor HDAC9 (histone deacetylase). Diseases associated with HDAC9 include gastrointestinal neuroendocrine tumor [72], medulloblastoma [73] and cutaneous squamous cell carcinoma [74]. The MSCs regulation seems to be associated with that molecule [75]. The histone deacetylases, in general, are known to induce/stabilize stemness phenotypes [76] and are associated with worse overall survival in chronic lymphocytic leukemia [77]. All these details might point towards the fact that the SP cells stabilize their presumed stemness character, amongst other mechanisms, by epigenetic regulatory circuits via HDAC9. The remarkable cluster of differential histone proteins might also contribute to this modified phenotype.

## 4. Conclusions

Taken together, the side population of the Ewing cell line CADO-ES1 (EWSR1-ERG fusion gene) shows a modification of AP-1–mediated transcriptional control and APC/c-CDC20-mediated cell cycle regulation. CDC20, besides other oncogenes and tumors suppressor genes, may play an oncogenic role in the development and progression of Ewing sarcoma of at least the given type. To sum up, the Ewing SP cells have their own biology apart from the main sarcoma cell population and their CSC properties might be imprinted in pathways centered around AP-1 and APC/c-CDC20. Stemness properties might be supported by the epigenetic modifier HDAC9 amongst others.

## 5. Materials and Methods

### 5.1. Cell Line and Cell Culture Conditions

The Ewing sarcoma cell line CADO-ES-1 was obtained from the Leibniz-Institute DSMZ (Braunschweig, Germany). It was cultivated in collagen-coated tissue culture flasks in RPMI-1640 medium (Sigma Aldrich, Taufkirchen, Germany) supplemented with 10% fetal calf serum (Gibco Life Technologies, Darmstadt, Germany), 2 mmol/L l-glutamine (Gibco) and an additional 1% of penicillin/streptomycin in a humidified atmosphere at 37 °C and 5% CO_2_. Microbiological analysis was carried out routinely using a PCR Mycoplasma Detection Kit (AppliChem GmbH, Darmstadt, Germany) according to the manufacturer’s instructions.

MSC cells were obtained from four Ewing sarcoma patients by density gradient centrifugation, resuspended in RPMI-1640 culture medium and frozen in liquid nitrogen or cultured for further use. All patients were included into the multicenter E.U.R.O Ewing 99 (EE99 NCT00020566, 12/02/1999) and Ewing 2008 (NCT00987636, 01/10/2009) trials. The multicenter studies were approved by the University of Muenster Ethical Board, and informed consent was obtained from patients and/or their legal guardians in accordance with the Declaration of Helsinki.

### 5.2. Side Population

The side population was established by staining the CADO-ES1 cells by ‘Hoechst 33342’ dye and a subsequent fluorescence-activated cell sorting (FACS). Staining procedure, analysis and sorting were done essentially as described in Reference [24]. In brief: CADO-ES1 (0.5–1 × 10^6^ cells/mL for side population analysis, 10 × 10^6^/mL for sorting experiments) were stained with 5 µg/mL Hoechst33342 for 90 min. at 37 °C in the presence and absence of 50 µM Verapamil. Unstained cells served as a negative control. Afterwards, cells were centrifuged and resuspended in 0.15 mL (for analysis) or 1–2 mL (for sorting) HBSS (Hanks balance salt solution) with 2% FCS and 10 mM HEPES buffer. In addition, unstained cells were resuspended in HBSS, 2% FCS, 10 mM HEPES and 2 µg/mL Propidum Iodide to exclude dead cells from flow cytometry analysis and fluorescence-activated cell sorting. Appendix A shows the living cells (gate R1) as well as the non-SP- (R2) and SP-gate (R3). Sorted cells were used for further experiments (see Section 5.3, Section 5.4 and Section 5.5) or pelleted, shock frozen in liquid nitrogen and stored at −80 °C.

### 5.3. Cytotoxiticity Assay

For the cell viability assays, 5000 CADO-ES1, SP or non-SP cells/well/100 µL were seeded in collagen-coated 96-well plates. Cells were allowed to adhere for 24 h before treatment. To determine growth inhibition relative to untreated controls, cells were treated with the drugs Etoposide, Doxorubicin and Cis-Platin at five different concentrations (0.01–100 µM). Each concentration was tested in four replicates. After 72 h of treatment, cells were incubated with MTT (3 (4.5 dimethylthiazol 2 yl) 2.5 diphenyltetrazolium bromide) for 3 h. Supernatant was discarded, cells lysed with 100 µL Iso-Propanol and the absorbance was measured in a plate reader at 550 nm. Each experiment was conducted in at least six independent biological replicates.

### 5.4. Colony Assy

Anchorage-independent growth was analyzed by plating 500 or 1000 CADO-ES1, SP or non-SP cells as a single-cell suspension in semisolid medium containing 0.9% (*w*/*v*) methylcellulose in IMDM, 15% FCS (*v*/*v*) and 0.5% (*w*/*v*) BSA. For each assay, 1 mL of the cell suspension was plated in duplicate into sterile 35 mm Petri dishes (Nunc, Karlsruhe, Germany) that were incubated at 37 °C. Colonies were counted after 11–14 days and each experiment was repeated at least two times.

### 5.5. In Vitro Differentiation Potential

The in vitro differentiation potential of CADO-ES1, SP- and non-SP-cells was analyzed by using the culture and adipogenic-differentiation conditions used for mesenchymal stem cells. In brief: When plated MSCs, sorted SP and non-SP cells reached near confluency (day 0), the medium was changed to Diff-Med 1 (RPMI-1640, 10% FCS, 1 µM Dexamethasone, 0.2 mM Indomethacine, 0.01 mg/mL Insulin and 0.5 mM IBMX (3-Isobutyl-1-methyl-xanthinie)) for 5 days. Then a second medium change to Diff-Med 2 (RPMI-1640, 10% FCS, 0.1 mg/mL Insulin) was executed for 2 days. These two medium changes were repeated a second time and finally cells were stained with OIL-RED-O solution (Sigma-Aldrich). OIL-RED-O is used to stain lipids within cells. Briefly, cells were fixed with formalin, washed with 50% ethanol and stained with OIL-RED-O solution for 20 min. After washing with 50% ethanol, water-stained cells were analyzed under a microscope.

### 5.6. RNA Isolation and Library Preparation

RNA isolation was performed based on FACS sorted and shock frozen cells (between 0.3 and 1.0 × 10^6^ cells) with TRIZOL reagent and the PureLink RNA isolation Mini Kit according to the manufactures protocol (Thermo Fisher Scientific, Waltham, MA, USA).

Total-RNA quantity was measured using a Nanodrop 2000 Spectrophotometer (Peqlab Biotechnologie GmbH, Erlangen, Germany); integrity was checked by an Agilent 2100 Bioanalyzer (Agilent Technologies Deutschland GmbH & Co. KG, Waldbronn, Germany). The ribosomal RNA was removed with RiboMinus Eukaryote Kit for RNA-Seq (Invitrogen, Darmstadt, Germany) following the manufacturer’s instructions.

Whole transcriptome libraries were prepared with a Solid Total RNA-Seq Kit (Life Technologies GmbH, Darmstadt, Germany). Sequencing was performed on a LifeTechnologies SOLiD4 for the first four samples (indexed and pooled) and, after a technology upgrade, on a LifeTechnologies SOLiD5500xl next-generation sequencer for the following eighteen samples (indexed and pooled). The read lengths on the SOLiD4 sequencer was 50/35 nucleotides and on the SOLiD5500xl sequencer 75/35 nucleotides. Sequencing chemistries on both sequencers were technically the same, i.e., Sequencing by Oligonucleotide Ligation and Detection (SOLiD). Ambion ERCC RNA Spike-In Control Mixes (Invitrogen) were spiked during library preparation for quality control and normalization during bioinformatics analysis following the supplier’s recommendations. The library preparation protocols for both sequencing runs, on the SOLiD4 and the SOLiD 5500xl, were nearly identical, with only a few minor differences, which are specified below.

In brief, rRNA-depleted total-RNA was fragmented by RNase III and purified using the RiboMinus Concentration Module (Invitrogen) samples. Afterwards, a quality check for quantity and fragment size was performed using a Qubit fluorometer (Life Technologies) and a Bioanalyzer (Agilent). The resulting fragment size ranged from 100 to 200 nucleotides. Adapters were added to the fragments using the hybridization master mix (SOLiD Total RNA-Seq Kit), followed by a reverse transcription. The SOLiD4-samples were purified with the MinElute PCR Purification Kit (Qiagen, Hilden, Germany) and the purified cDNA were size-selected during electrophoresis with the Novex TBE-Urea Gel 10% (Invitrogen) at 180 V for 25 min. The resulting fragment size ranged from 150 to 250 nucleotides.

The SOLiD 5500xl-samples were size-selected using the Agencourt AMPureXP Reagent (Beckman Coulter GmbH, Krefeld, Germany) by performing two rounds of bead capture, wash and elution to ensure complete capture and size-selection of the desired cDNA (fragment size: 100–150 nucleotides).

The purified fragments of desired size were used as templates for the following amplification. During the PCR amplification, unique SOLiD RNA sample barcodes were added to allow a pooled sequencing of the 4 and 18 samples, respectively. The PCR reactions were performed at 95 °C for 5 min and then cycled at 95 °C for 30 s, 62 °C for 30 s and 72 °C for 30 s for 15 cycles and a final elongation at 72 °C for 7 min. A final clean-up was performed using the Invitrogen PureLink PCR Micro Kit (Invitrogen).

Final quality control for size distribution and yield of the PCR products was done by a Bioanalyzer (Agilent) and a Nanodrop 2000 (Peqlab). The Bioanalyzer smear peaks ranged from 25 to 200 nucleotides. Emulsion-PCR and bead enrichment were performed using the SOLiD EZ Bead System (Life Technologies). A workflow analysis run was performed for the SOLiD4 run to verify the quality and density of the template beads. Approximately 120 million enriched beads for each sample were deposited on the sequencing slide. Finally, the libraries were sequenced, resulting in color-space reads as an output format. The whole data set comprising the presented data is available online: E-MTAB-6067 [78].

### 5.7. Raw Data Processing/Controls

The SOLiD color-space reads are checked for the presence of primer sequences and are trimmed using the tool Cutadapt (Martin, 2011). The reads were then mapped with TopHat aligner in ‘transcriptome + genome’ mapping mode to the reference transcriptome. The reads that did not find an alignment (unmapped reads) has a second chance to be mapped to the genomic regions [79]. The aligner is supplied with human genome reference (GRCh38/hg38 patch 10) and the corresponding Ensembl transcriptomic GTF annotation [80]. A Python stand-alone script HTSseq-count was used in ‘intersection-strict’ mode to obtain gene-level read counts. The script is supplied with the aligned sequencing reads and the Ensemble GTF annotations are used as inputs [81]. The obtained counts for each library were consolidated into a single expression count matrix to facilitate descriptive analysis and differential expression calling.

### 5.8. Differential Analysis/Controls

DESeq2 is a method for differential analysis of count data, and is implemented as an R Bioconductor package DESeq2 [82]. As the DESeq2 model internally corrects for the library size, it is supplied with un-normalized count data as an input to identify significant changes in gene expression between SPs and non-SPs. The FDR cut-off criteria for the DEGs are set to a default value of 0.1, because the set size plays a critical role in the enrichment analyses. Therefore, ranking is important, but less so if a single candidate is finally rock solid. For all the downstream analyses a value of 0.05 is in effect.

### 5.9. Functional and Pathway Enrichment Analysis

The Gene Ontology (GO) and KEGG (Kyoto Encyclopedia of Genes and Genomes) provide controlled vocabulary for the classification and annotation of genes and gene products in terms of their putative functions and pathways [83,84]. The functional annotations are categorized as ‘Biological process’, ‘Molecular function’ and ‘Cellular component’.

Biological Networks Gene Ontology tool (BiNGO), a Cytoscape plugin, has been used to identify the GO terms that are statistically overrepresented in a set of DEGs [85,86].

The Database for Annotation, Visualization, and Integrated Discovery tool (DAVID), was used to identify the significantly enriched KEGG pathways (Dennis et al., 2003). REACTOME, a peer-reviewed knowledgebase of biomolecular pathways has been used to obtain in-depth information about the enriched pathways both in view of their hierarchy and the metabolic processes involved [87]. The Benjamini and Hochberg multiple testing correction (FDR) was applied throughout to filter the significantly associated functions and pathways [88].

### 5.10. Identification of Tumor-Associated Genes

The TSGene database is a knowledge-base of curated tumor suppressor genes from multiple cancer types [89]. ONGene is a database of curated oncogenes that are involved in the initiation and promotion of cancer progression [90].

With the aim of finding DEGs associated with tumors, a comprehensive list of tumor suppressor genes and oncogenes was created by combining the information obtained from TSGene [91] and ONGene [92] databases. The genes in this list are annotated into three categories namely, ‘oncogene’ for the genes involved in oncogenesis, ‘tumor suppressor gene’ for the genes involved in tumor suppression and ‘both’ for the genes involved in both oncogenesis and tumor suppression. The list created by the aforementioned procedure contains 674 oncogenes, 1088 tumor suppressor genes and 129 genes falling into both the categories. This list has been used to annotate the DEGs with tumor-associated properties.

### 5.11. Construction of PPI Network and the Subnetwork Analysis

STRING (Search Tool for the Retrieval of Interacting Genes) is a database that contains consolidated information of both known and predicted PPI data for a large number of organisms and experiments [93,94]. The DEGs were used as an input to the STRING database to build the interaction network of the gene products. The minimum interaction score is set to the default value of 0.4. The results were exported and imported into Cytoscape, a software environment for visualization and integration of biomolecular interaction data [85]. The network and subnetwork analysis was also done in Cytoscape. Subnetwork analysis denotes the identification of sets of genes and interactions that participate in a meaningful biological function that are ‘active subnetworks/modules’ [95]. Among a wide range of available methods for mining the active modules, jActiveModules has been chosen for its precision and ease of use [96]. jActiveModules is available as a Cytoscape plugin. It screens the molecular interaction network to identify the expression-activated subnetworks. It operates by scoring the subnetworks and implementing one of the two available search algorithms to identify top-scoring subnetworks [27]. The analysis was conducted with a greedy search algorithm due to its precision over the simulated annealing algorithm [96].

## Figures and Tables

**Figure 1 ijms-19-03908-f001:**
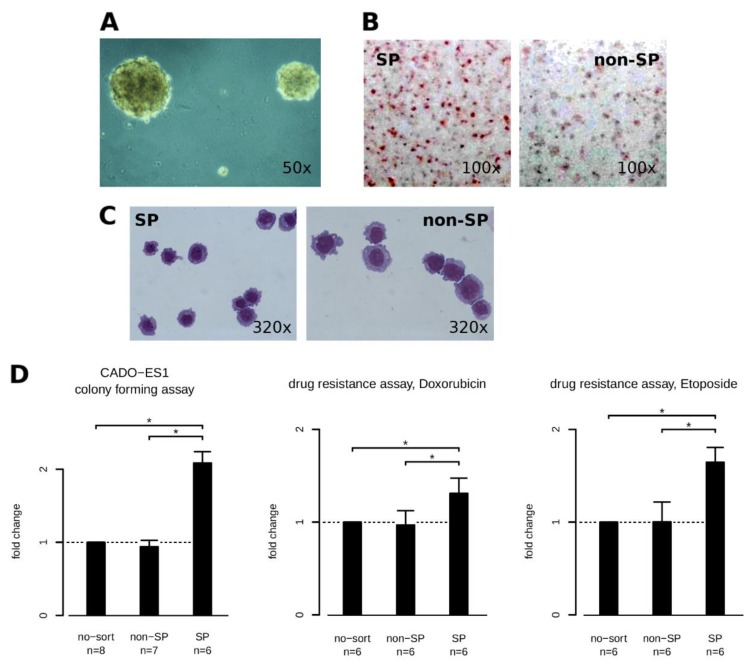
Stemness features of the side population. In image (**A**) respective panel (**D** the left drawing), the colony-forming potential of SP compared to non-SP cells is illustrated. The image shows representative colonies. Panel (**B**) shows OIL-RED-O-stained cells after adipogenic differentiation. More dark red cells indicate more adipogeneic differentiation. The slightly smaller SP cells are seen in Panel (**C**) (May-Grünwald-Giemsa staining). In panel (**D**), the two rightmost diagrams show the higher resistance of SP cells compared to non-SP cells (reference no-sort) after treatment with cytotoxic drugs for 72 h. Doxorubicine was effective at 0.1 µM while Etoposide is shown at 1 µM. A star indicates a significant difference using the t test and an alpha error of 5%. ‘*n*’ denotes the number of biological replicates behind a condition. Horizontal brackets with a star indicate a significant difference between bars (alpha < 0.05).

**Figure 2 ijms-19-03908-f002:**
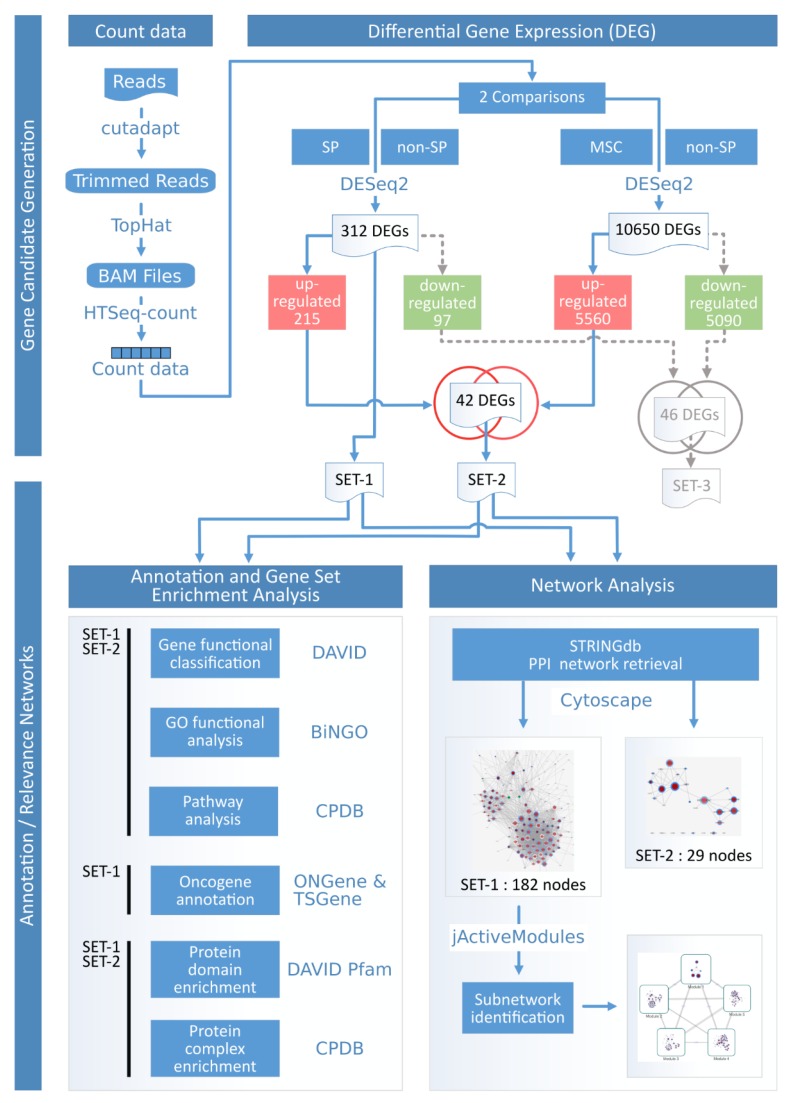
Workflow of the data analysis. This graph explains how the different gene sets are created and in which type of analysis they are utilized. The starting point is the top left.

**Figure 3 ijms-19-03908-f003:**
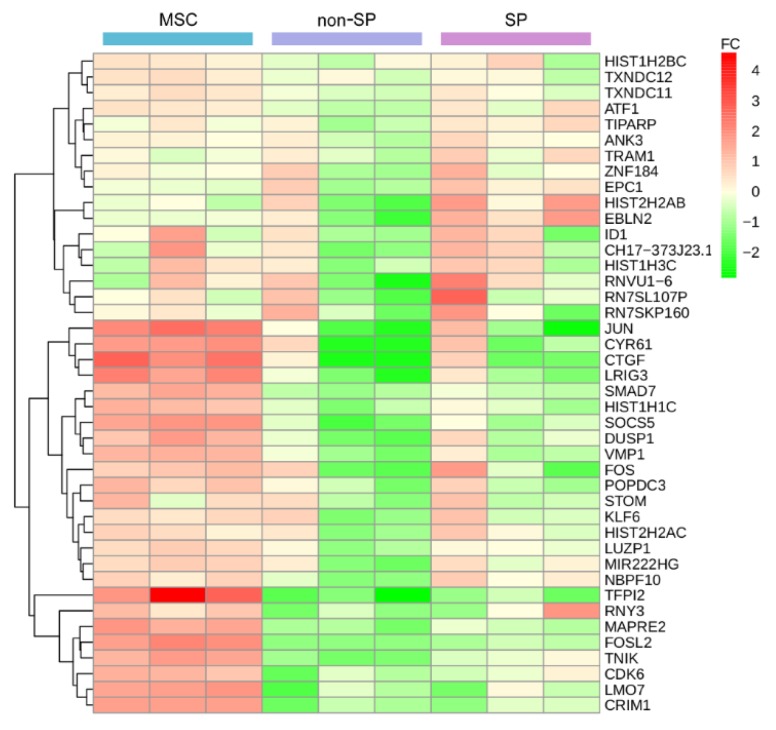
Heatmap of the counts for 42 up-regulated genes from SET-2. The counts are quantile normalized (preprocessCore) and rlog transformed (DESeq2) prior to plotting. The color bar on the right is defining the fold change values (FC) concerning gene average. Green colors denote levels of down-regulation, while red colors indicate levels of up-regulation. The order of SP and non-SP experiments reflects their paired nature (1,2,3,1,2,3). The hierarchical cluster tree on the left is based on the Euclidean measure.

**Figure 4 ijms-19-03908-f004:**
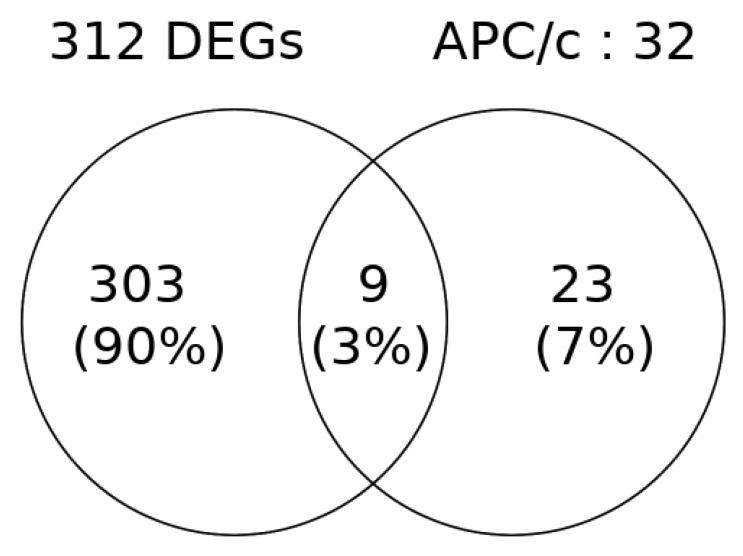
Intersecting number of genes between the SET-1 DEGs and the member of the pathway involved in the Activation of APC/c and CDC20-mediated degradation of mitotic proteins. The nine intersecting members include PLK1, CDK1, MAD2L1, BUB1B, UBE2C, CCNA2, CDC20, NEK2, and CCNB1.

**Figure 5 ijms-19-03908-f005:**
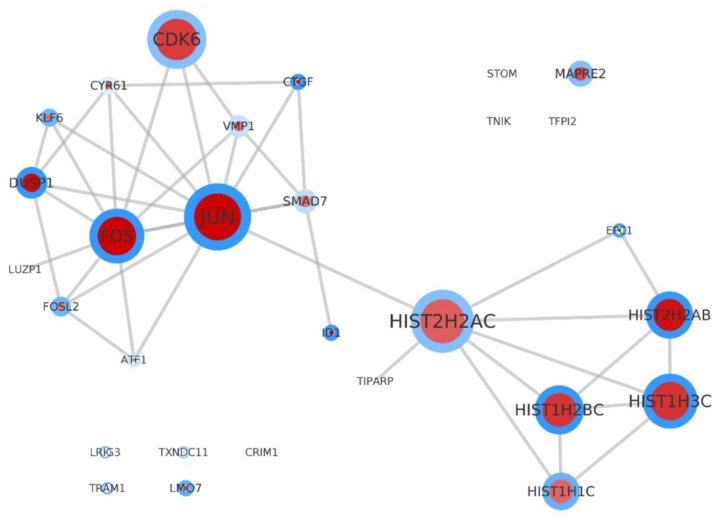
Subset of the PPI relevance network with the genes from SET-2. The gene products are represented by circles and their interactions are represented by edges. The size of the circles indicate the degree of connectivity to other partners. The larger the circle, the greater the degree. Red circles represent the products of up-regulated DEGs and green circles represent the products of down-regulated DEGs. The intensity of the colors corresponds to the log2 fold changes. The higher the fold change, the higher the color intensity. The blue color around the circles represents the *p*-value. The lower the *p*-value, the higher the color intensity. The PPI is underlining the relevance of AP-1 and the histone cluster.

**Figure 6 ijms-19-03908-f006:**
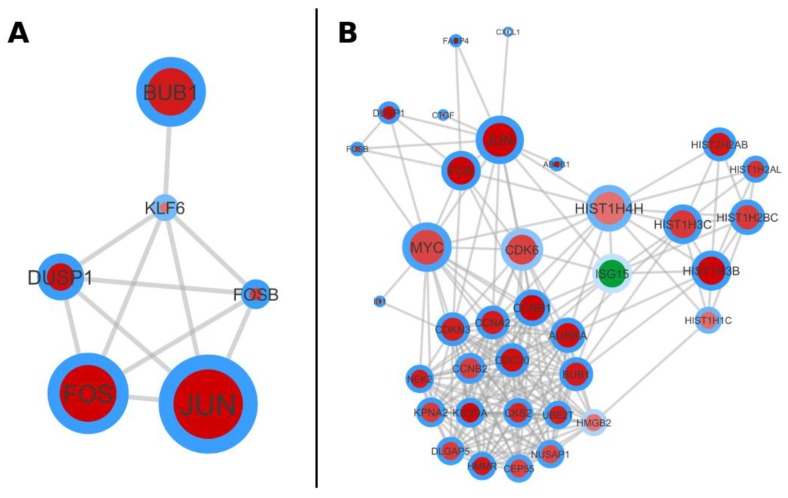
Highest scoring subnetworks of SET-1. (**A**) shows the first highest scoring subnetwork with six nodes (Module 1, see Appendix A). (**B**) shows the second highest scoring subnetwork with 36 nodes (Module 2).

**Table 1 ijms-19-03908-t001:** Significantly enriched Biological Processes in SET-1 using BiNGO. The third column displays the number of genes from SET-1 enriched in the respective biological process. The last column of the table displays the number of genes from the input list present in at least one pathway. To obtain this result, the third column of the current table is fed as an input to the CPDB analysis (cf. Table 2). The *p* values are FDR (false discovery rate) corrected.

GO-ID	GO Name	Gene Counts	*p* values (FDR)	Candidates Also Part of Enriched Pathways
22403	cell cycle phase	48	2.3 × 10^−31^	36 (75%)
279	M phase	44	3.1 × 10^−31^	33 (75%)
6996	organelle organization	75	3.1 × 10^−31^	65 (87%)
22402	cell cycle process	52	2.3 × 10^−30^	39 (75%)
278	mitotic cell cycle	44	5.6 × 10^−30^	34 (77%)
87	M phase of mitotic cell cycle	36	3.5 × 10^−28^	27 (75%)
280	nuclear division	35	1.8 × 10^−27^	26 (74%)
7067	mitosis	35	1.8 × 10^−27^	26 (74%)
7049	cell cycle	55	5.1 × 10^−27^	42 (76%)
48285	organelle fission	35	5.5 × 10^−27^	26 (74%)

**Table 2 ijms-19-03908-t002:** Enriched pathway-based sets for SET-1 and SET-2 as obtained from the CPDB analysis.

Pathway Name	Set Size	Candidates Contained	*p* Value	*q* Value	Source
**SET-1**					
Cell Cycle, Mitotic	468	53 (11%)	1.7 × 10^−39^	1.0 × 10^−36^	Reactome
Cell Cycle	551	55 (10%)	4.0 × 10^−38^	1.2 × 10^−35^	Reactome
M Phase	267	40 (15%)	3.0 × 10^−34^	6.1 × 10^−32^	Reactome
RHO GTPase Effectors	299	32 (11%)	1.2 × 10^−22^	1.9 × 10^−20^	Reactome
Condensation of Prophase Chromosomes	77	20 (26%)	1.9 × 10^−22^	2.3 × 10^−20^	Reactome
Mitotic Prophase	143	24 (17%)	7.9 × 10^−22^	8.0 × 10^−20^	Reactome
Senescence-Associated Secretory Phenotype (SASP)	113	22 (20%)	1.4 × 10^−21^	1.2 × 10^−19^	Reactome
Cellular Senescence	192	25 (13%)	6.9 × 10^−20^	5.2 × 10^−18^	Reactome
HATs acetylate histones	143	22 (15%)	3.8 × 10^−19^	2.6 × 10^−17^	Reactome
HDACs deacetylate histones	94	19 (20%)	5.3 × 10^−19^	3.2 × 10^−17^	Reactome
**SET-2**					
Cellular Senescence	192	9 (5%)	1.8 × 10^−10^	4.2 × 10^−08^	Reactome
Oxidative Stress-Induced Senescence	129	7 (6%)	9.1 × 10^−09^	1.1 × 10^−06^	Reactome
Cellular responses to stress	393	9 (2%)	9.6 × 10^−08^	7.4 × 10^−06^	Reactome
Senescence-Associated Secretory Phenotype (SASP)	113	6 (5%)	1.4 × 10^−07^	7.8 × 10^−06^	Reactome
AP-1 transcription factor network	71	5 (7%)	4.3 × 10^−07^	2.0 × 10^−05^	PID
HATs acetylate histones	143	5 (4%)	1.4 × 10^−05^	5.3 × 10^−04^	Reactome
MAPK targets/Nuclear events mediated by MAP kinases	31	3 (10%)	4.2 × 10^−05^	1.4 × 10^−03^	Reactome
HDACs deacetylate histones	94	4 (4%)	5.1 × 10^−05^	1.5 × 10^−03^	Reactome
ErbB1 downstream signaling	107	4 (4%)	8.2 × 10^−05^	2.1 × 10^−03^	PID
Hfe effect on hepcidin production	7	2 (27%)	9.9 × 10^−05^	2.1 × 10^−03^	Wiki-pathways

**Table 3 ijms-19-03908-t003:** Top 10 enriched protein domains in up-regulated genes of SET-1 and all the enriched protein domains of SET-2 as obtained from DAVID according to the PFAM database. The reported *p* values are FDR corrected.

Term	Protein Domains	Candidates Contained	*p* Value (FDR)
**SET-1**		set size 204	
PF00125	Core histone H2A/H2B/H3/H4	18 (9%)	9.8 × 10^−18^
PF00225	Kinesin motor domain	6 (3%)	6.1 × 10^−03^
PF00170	bZIP transcription factor	5 (2%)	8.4 × 10^−03^
PF00069	Protein kinase domain	11 (5%)	9.9 × 10^−02^
PF02984	Cyclin, C-terminal domain	3 (1%)	3.8 × 10^−01^
PF00219	Insulin-like growth factor-binding protein	3 (1%)	3.8 × 10^−01^
PF08311	Mad3/BUB1 homology region 1	2 (1%)	4.0 × 10^−01^
PF00307	Calponin homology (CH) domain	4 (2%)	4.6 × 10^−01^
PF00093	von Willebrand factor type C domain	3 (1%)	4.6 × 10^−01^
PF00134	Cyclin, N-terminal domain	3 (1%)	5.5 × 10^−01^
**SET-2**		set size 41	
PF00170	bZIP transcription factor	4 (10%)	8.1 × 10^−04^
PF00125	Core histone H2A/H2B/H3/H4	4 (10%)	5.9 × 10^−03^
PF00219	Insulin-like growth factor binding protein	3 (7%)	9.2 × 10^−03^
PF00093	von Willebrand factor type C domain	3 (7%)	1.4 × 10^−02^
PF00007	Cystine-knot domain	2 (5%)	2.4 × 10^−01^

**Table 4 ijms-19-03908-t004:** Enriched protein complex-based sets for SET-1 and SET-2 as obtained from the CPDB database.

Name of Protein Complex	Set Size	Candidates Contained	*p* Value	*q* Value	Source
**SET-1**					
AP-1	5	4 (80%)	1.4 × 10^−07^	1.5 × 10^−05^	INOH
CycB-Cdc2.complex	3	3 (100%)	2.2 × 10^−06^	1.2 × 10^−04^	Spike
Centrosome:AURKA:TPX2:HMMR	75	8 (11%)	4.8 × 10^−06^	1.3 × 10^−04^	Reactome
MASH1 promoter-coactivator complex	37	6 (16%)	7.2 × 10^−06^	1.3 × 10^−04^	CORUM
Nek2A:MCC:APC/C complex	22	5 (23%)	7.6 × 10^−06^	1.3 × 10^−04^	Reactome
H3.1 com	38	6 (16%)	8.5 × 10^−06^	1.3 × 10^−04^	PINdb
hBUBR1:hBUB3:MAD2*:CDC20 complex	4	3 (75%)	8.6 × 10^−06^	1.3 × 10^−04^	Reactome
Cell cycle kinase complex CDC2	6	3 (50%)	4.2 × 10^−05^	5.7 × 10^−04^	CORUM
Histone H3.1 complex	32	5 (16%)	5.3 × 10^−05^	6.4 × 10^−04^	CORUM
Emerin regulatory complex	18	4 (22%)	7.3 × 10^−05^	7.3 × 10^−04^	CORUM
**SET-2**					
AP-1	5	3 (60%)	1.4 × 10^−07^	2.7 × 10^−06^	INOH
p-2S-cJUN:p-2S,2T-cFOS	2	2 (100%)	6.0 × 10^−06^	2.4 × 10^−05^	Reactome
Fra2/JUN	2	2 (100%)	6.0 × 10^−06^	2.4 × 10^−05^	PID
c-FOS/c-JUN/CREB/CREB	3	2 (67%)	1.8 × 10^−05^	2.4 × 10^−05^	BioCarta
ERG-JUN-FOS DNA-protein complex	3	2 (67%)	1.8 × 10^−05^	2.4 × 10^−05^	CORUM
JUN/FOS/ER alpha	3	2 (67%)	1.8 × 10^−05^	2.4 × 10^−05^	PID
ETS2-FOS-JUN complex	3	2 (67%)	1.8 × 10^−05^	2.4 × 10^−05^	CORUM
JUN/FOS/GATA2	3	2 (67%)	1.8 × 10^−05^	2.4 × 10^−05^	PID
cortisol/GR alpha (monomer)/JUN/FOS	3	2 (67%)	1.8 × 10^−05^	2.4 × 10^−05^	PID
p-2S-JUN:p-2S,2T-FOS:IGFBP7 Gene	3	2 (67%)	1.8 × 10^−05^	2.4 × 10^−05^	Reactom

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
