# Peer review of "Defining a Characteristic Gene Expression Set Responsible for Cancer Stem Cell-Like Features in a Sub-Population of Ewing Sarcoma Cells CADO-ES1"

_ijms, 2018, doi:10.3390/ijms19123908_

Reviewer 1 Report

    The manuscript entitled “Defining a Characteristic Gene Expression Set Responsible for Cancer Stem Cell-Like Features in a sub-population of Ewing Sarcoma Cells” by Hotfilder et al. is a study of cancer stem cell-like cells using Ewing sarcoma cell line CADO-ES1. The authors isolated stem cell-like cells (sub-population cells) from Ewing sarcoma cells and found that the Ewing sub-population cells had their own biology apart from the main sarcoma cell population and the mesenchymal stem cells. The authors concluded that AP1 and APC-CDC20 played an oncogenic role in the development and progression of Ewing sarcoma because an altered expression structure of sub-population cells centered on the AP-1 and APC/c complex. 

     This manuscript is well-organized and grounded in scientific experiments. This article will be of interest to readers of International Journal of Molecular Sciences. I have a few comments on this manuscript.

Comments

1. My biggest concern is reproducibility of the results of this article. The authors used only one cell line of Ewing sarcoma despite the existence of several Ewing sarcoma cell lines which are available to the public. The authors are encouraged to use additional cell lines to confirm these results.

2. The authors are encouraged to add some basic studies to conclude that AP1 and APC-CDC20 played an oncogenic role in the development and progression of Ewing sarcoma.

Author Response

Manuscript ID: ijms-384699

Defining a Characteristic Gene Expression Set Responsible for Cancer Stem Cell Like Features in a sub-population of Ewing Sarcoma Cells CADO-ES1

Response to reviewer  ( 1 )

we appreciate the chance to improve the given manuscript and we also thank the two anonymous reviewers for the comprehensive advice which helped us to improve the manuscript.

According to reviewer 2 the title was slightly adjusted.

All the concerns of reviewer 1 [given here in italics] were addressed [our answers below each section].

As far as necessary the corresponding corrections were also included into the main manuscript.

Comments and suggestions for authors

The manuscript entitled “Defining a Characteristic Gene Expression Set Responsible for Cancer Stem Cell-Like Features in a sub-population of Ewing Sarcoma Cells” by Hotfilder et al. is a study of cancer stem cell-like cells using Ewing sarcoma cell line CADO-ES1. The authors isolated stem cell-like cells (sub-population cells) from Ewing sarcoma cells and found that the Ewing sub-population cells had their own biology apart from the main sarcoma cell population and the mesenchymal stem cells. The authors concluded that AP1 and APC-CDC20 played an oncogenic role in the development and progression of Ewing sarcoma because an altered expression structure of sub-population cells centered on the AP-1 and APC/c complex.

This manuscript is well-organized and grounded in scientific experiments. This article will be of interest to readers of International Journal of Molecular Sciences. I have a few comments on this manuscript.

We are delighted to hear that.

Comments

1. My biggest concern is reproducibility of the results of this article. The authors used only one cell line of Ewing sarcoma despite the existence of several Ewing sarcoma cell lines which are available to the public. The authors are encouraged to use additional cell lines to confirm these results.

The experiments are based on true sample triplicates in the case of the CADO-ES1 and on 4 biological replicates in the case of MSCs. The analytical procedures were based on reliable p value criteria with FDR correction. The well recognized R package DEseq2 was applied for the differential analysis which considers a lot of important determining factors like, NGS library normalization, variance/dispersion estimates, and sample batch effects. So the conclusions for the CADO-ES1 cell line is reasonable stable, and together with the consistent a priory information out of the databases, empirically sound.

Certainly we agree, that it would be preferable to see other stable Ewing cell lines in this experimental design. But two primary Ewing cell line cultures, which we tested, were not reasonably stable.

Concerning the stability of the results, the main objective was not to map subgroup differences in Ewing sarcoma nor to characterize phenotypic subgroup differences on the molecular pathway level, instead to characterize very basic tumor pathways responsible for tumor maintenance and putatively susceptible for tumor progression. For that, the generated data quality and the resulting observations form a consistent and reasonably robust outcome. So it is unlikely, that the presented results become void. Furthermore, these basic pathways should be valid for the whole tumor entity and even beyond. So the presented data is a good resource for further and extended studies.

In conclusion, this study contributes a comprehensive and self-contained insight into a Ewing side population with stemness features, a valid, free, and comprehensive data set, and is therefore forming a stable basis to create further research efforts.

2. The authors are encouraged to add some basic studies to conclude that AP1 and APC-CDC20 played an oncogenic role in the development and progression of Ewing sarcoma.

In the field of Ewing sarcoma these molecular factors were not really considered in publications so far either alone or in combination. The publication of Satterfield et al. (2017) is touching this topic in combination with microRNAs and Whitehurst et al. (2008) is contributing somewhat. In cancer in general several publications are out, which are pointing to the 'hub' functionality of these factors in tumorgenesis. AP-1 and APC/c are often observed to act in conjunction with other factors directly linked to the molecule complexes or as downstream targets like CDC20 or CDH1, CCND1, CDKN2A, UbcH10 or BUB1/BUBR1, to name some. The combination of epigenetic factors, microRNAs and gene expression is truly the combination which will expose regulatory circuits in more detail. Nevertheless, the gene expression pattern in conjunction with the protein expression (if measured), is the first and basic step to decipher differences in the regulatory system of the cell, because it is the outcome of regulation.

We added a comprehensive section in the discussion part of the manuscript to enhance the existing argumentation and to improve the support for the conclusions.

Reviewer 2 Report

To more effectively and specifically target malignant cells of a cancer two important questions must be addressed: which original lineage the malignant cells come from and what causes the transformation? The origin of Ewing sarcoma has been a subject of debate. It was thought to be from primitive neuroectodermal cells, but now many believe that it arises from a mesenchymal stem cell (MSC). Based on RNA deep sequencing data and various bioinformatic analyses Hotfilder et al intended to compare a side population (SP) of the Ewing sarcoma cell line used in the study to the non-SP and to the MSC isolated from 4 Ewing sarcoma patients.  They desired to find out the active regulatory mechanisms underlying the cancer stem cell (CSC) properties of SP cells in Ewing’s sarcoma.  Upon reading through the story however, I feel that the original design of this study is seemingly to see if CSC-like population can be isolated from the sole Ewing model cell ling CADO-ES1 and whether this SP resembles MSC isolated from Ewing sarcoma patients in terms of sharing the same regulatory network as compared to the Non-SP.  I have no question about the technologies and methods used in the study to analyze the data, but I do have some major concerns about the cells used and experimental design they applied.

Major concerns:

1.       I am skeptical about the results and conclusions obtained from only one Ewing sarcoma cell line used in the study.

2.       It is still under debate whether Ewing sarcoma cells originate from MSC; it is a better idea to directly analyze SP vs MSC.  Instead, authors analyzed MSC vs Non-SP.

3.       Since the used CADO-ES1 cell line is so different from the used MSC, the reader would easily conclude that Ewing sarcoma cells do not originate from the MSC no matter whether analyze SP vs MSC or MSC vs Non-SP.  I don’t know why authors did not say anything about this in the results and in the discussion.

4.       The bioinformatic analyses demonstrated that the enhanced cell cycle activities underline the self-renewal of the SP.  The author should also emphasize that changes in the expression of epigenetic modifiers likely underline the stemness properties of the SP.

Minor concerns:

1.       All abbreviations need be spelt out when first appear in the text like ES, FDR, TSG, PPI, etc.

2.       Lines 98 – 99, What do you mean 'better"? More SP cells stained? Then how many more?

3.       Line 100, “more compact morphology” is not accurate.  Based on figure 1C, SP cell size is smaller than non-SP, but the morphology basically is the same.

4.       Line 105, figure 1, the scales need be presented directly in the pictures.

5.       Figure 1D, what the ‘N’s mean?

Author Response

Manuscript ID: ijms-384699

Defining a Characteristic Gene Expression Set Responsible for Cancer Stem Cell Like Features in a sub-population of Ewing Sarcoma Cells CADO-ES1

Response to reviewer  ( 2 )

we appreciate the chance to improve the given manuscript and we also thank the two anonymous reviewers for the comprehensive advice which helped us to improve the manuscript.

According to reviewer 2 the title was slightly adjusted.

All the concerns of reviewer 2 [given here in italics] were addressed [our answers below each section].

As far as necessary the corresponding corrections were also included into the main manuscript.

Comments and suggestions for authors

To more effectively and specifically target malignant cells of a cancer two important questions must be addressed: which original lineage the malignant cells come from and what causes the transformation? The origin of Ewing sarcoma has been a subject of debate. It was thought to be from primitive neuroectodermal cells, but now many believe that it arises from a mesenchymal stem cell (MSC). Based on RNA deep sequencing data and various bioinformatic analyses Hotfilder et al intended to compare a side population (SP) of the Ewing sarcoma cell line used in the study to the non-SP and to the MSC isolated from 4 Ewing sarcoma patients.  They desired to find out the active regulatory mechanisms underlying the cancer stem cell (CSC) properties of SP cells in Ewing’s sarcoma.  Upon reading through the story however, I feel that the original design of this study is seemingly to see if CSC-like population can be isolated from the sole Ewing model cell ling CADO-ES1 and whether this SP resembles MSC isolated from Ewing sarcoma patients in terms of sharing the same regulatory network as compared to the Non-SP.  I have no question about the technologies and methods used in the study to analyze the data, but I do have some major concerns about the cells used and experimental design they applied.

The title of the study is indeed slightly too apodictic. Therefore we changed the title to become more specific, less generalizing, which is better aligned with the core of the presented results.

Nevertheless it needs to be stated, that the power of the analysis is well suited to generate reliable results, which are robust pointers to affected biological pathways and basic mechanisms.

The experimental design of a double differential comparison is a) well established and b) a necessary basic element of generating the presented specific results. There is no other option, based on this data type (gene expression), to do so.

The cell line is ubiquitously used in the Ewing sarcoma research community and therefore owns some standardization features and enables the comparability to other results. The experimental approach needs some stability and less established primary cell cultures tend to own more sources of variability. So our decision to use this cell line CADO-ES1 and the cell culture model was primarily to decipher, on a very robust level, basic mechanisms, which trigger core decisions in the biological network of an Ewing driver cell population.

For this purpose the approach is well chosen.

Major concerns:

1.       I am skeptical about the results and conclusions obtained from only one Ewing sarcoma cell line used in the study.

This concern is valid, if, and only if, the divers of the sub-phenotypes of Ewing sarcoma cell lines or entities define the objectives. This is not the case here, where basic tumor and tumor progression pathways should be identified.

Indeed this is a case study and not a broad screening study, but the objective of this study, to identify core mechanistic details and sub-networks which are specific for a Ewing sarcoma side population with CSC features, can be approached well with this type of study. There are many tumor phenotypes but definitely less core mechanisms translating into cancer, cancer progression and CSCs.

2.       It is still under debate whether Ewing sarcoma cells originate from MSC; it is a better idea to directly analyze SP vs MSC.  Instead, authors analyzed MSC vs Non-SP.

The reviewer is right, the cell of origin is still under debate, but this is something where this study might also partly contribute to. The publication of Tirode et al. and some new ones, which we cite, is supporting our decision. If the study results really get sunk into the noise level of an expression analysis, they get as a whole unspecific and random. A conceptional interpretation based on annotation data will fail. This is not the case in the presented approach, based on a broad amount of a priory knowledge collected in the used databases and used for creating the results of this study.

Why is a linked double differential analysis of gene expression data better than a direct differential analysis? a) Combinatorially you raise specificity by using more conditions. b) The difference between MSCs and other cells is high. In our case 5K up-regulated genes are different. This is also the case with SPs. Only shrinking or selecting result sets by fold change and/or p value order will dilute the specific results in a lot of stable and non-interesting differences - so not advised. c) The SP population, as mentioned, is a fraction of the non-SP population. The intention is not to hock on non-SP features, instead to solely those ones which are SP features. Hence, non-SP features need to be sorted out.

The analysis design presented in the manuscript is therefore based on a well considered concept.

3.       Since the used CADO-ES1 cell line is so different from the used MSC, the reader would easily conclude that Ewing sarcoma cells do not originate from the MSC no matter whether analyze SP vs MSC or MSC vs Non-SP.  I don’t know why authors did not say anything about this in the results and in the discussion.

It is well established knowledge, that stem cells in general own complete different expression patterns, compared to more differentiated cell types. We apologize for not considering research communities which are not familiar with these observations.

We added a descriptive sentence in the results section.

4.       The bioinformatic analyses demonstrated that the enhanced cell cycle activities underline the self-renewal of the SP.  The author should also emphasize that changes in the expression of epigenetic modifiers likely underline the stemness properties of the SP.

Truly, the combination of epigenetic events, microRNA action, and presumably further molecular effectors form this distorted biological network of Ewing sarcoma and the defined SP cell population. The overlap of our differential sets with epigenetic modifiers in the dbEM database does result in one hit. We therefore drafted tow additional sections, considering this aspect, at the end of the discussion and added a small section in the results.

We thank the reviewer for this informative comment.

Minor concerns:

1.       All abbreviations need be spelt out when first appear in the text like ES, FDR, TSG, PPI, etc.

The manuscript was carefully revised, to intercept all these issues.

2.       Lines 98 – 99, What do you mean 'better"? More SP cells stained? Then how many more?

The expression “… better than …” has been replaced by “ … differentiate to  a higher extend into adipogenic cells compared to …”

3.       Line 100, “more compact morphology” is not accurate.  Based on figure 1C, SP cell size is smaller than non-SP, but the morphology basically is the same.

The description of SP as “more compact morphology” has been changed to “smaller”. This was also changed in the legend of Figure 1.

4.       Line 105, figure 1, the scales need be presented directly in the pictures.

The magnification of the pictures has been inserted to the lower right corner.

5.       Figure 1D, what the ‘N’s mean?

'n' in these plot types means number of biological replicates behind each bar graph.

We adjusted the Figure 1 legend accordingly.

p { margin-bottom: 6.25px; ; }a:link { }

Round  2

Reviewer 1 Report

   The revised manuscript entitled “Defining a Characteristic Gene Expression Set Responsible for Cancer Stem Cell Like Features in a sub-population of Ewing Sarcoma Cells CADO-ES1” by Hotfilder et al. is research on cancer stem cell-like features using Ewing sarcoma cell line CADO-ES1. The authors isolated stem cell-like cells (a sub-population of Ewing sarcoma cells) and found that those cells had their original biology apart from the main sarcoma cell population and the mesenchymal stem cells. The authors concluded that AP1 and APC-CDC20 played an oncogenic role in the development and progression of Ewing sarcoma because an altered expression structure of stem cell-like cells centered on the AP-1 and APC/c complex. 

   This manuscript is well-organized and grounded in scientific experiments. This article will be of interest to readers of International Journal of Molecular Sciences.

     The authors have revised their original manuscript partly according to the reviewers’ comments. In some points, the authors decided to keep their contents unchanged, however, their rebuttal seems almost reasonable.

Reviewer 2 Report

The authors have adequately addressed my concerns, now I agree to publish their manuscript without further revision.